# REALTIME QA: What's the Answer Right Now?

**Jungo Kasai**[♡*] **Keisuke Sakaguchi**[♣□*] **Yoichi Takahashi**[♣] **Ronan Le Bras**[◇]
**Akari Asai**[♠] **Xinyan Velocity Yu**[♥] **Dragomir Radev**[♦]
**Noah A. Smith**[◇♠] **Yejin Choi**[◇♠] **Kentaro Inui**[△♣□]

[♡]Toyota Technological Institute at Chicago [♣]Tohoku University [□]RIKEN [◇]Allen Institute for AI
[♠]University of Washington [♥]University of Southern California [♦]Yale University [△]MBZUAI

REALTIME QA realtimeqa.nlp@gmail.com @realtimeqa

## Abstract

We introduce REALTIME QA, a dynamic question answering (QA) platform that announces questions and evaluates systems on a regular basis (weekly in this version). REALTIME QA inquires about the *current* world, and QA systems need to answer questions about novel events or information. It therefore challenges static, conventional assumptions in open-domain QA datasets and pursues instantaneous applications. We build strong baseline models upon large pretrained language models, including GPT-3 and T5. Our benchmark is an ongoing effort, and this paper presents real-time evaluation results over the past year. Our experimental results show that GPT-3 can often properly update its generation results, based on newly-retrieved documents, highlighting the importance of up-to-date information retrieval. Nonetheless, we find that GPT-3 tends to return *outdated* answers when retrieved documents do not provide sufficient information to find an answer. This suggests an important avenue for future research: can an open-domain QA system identify such unanswerable cases and communicate with the user or even the retrieval module to modify the retrieval results? We hope that REALTIME QA will spur progress in instantaneous applications of question answering and beyond.[2]

## 1 Introduction

*How many home runs has Shohei Ohtani hit so far this season?* A user of a question answering (QA) system might ask such time-sensitive questions and seek out answers in *real time*. Widely-used evaluation benchmarks of QA systems, however, implicitly assume that answers are static regardless of the time of inquiry. Several recent works (Jia et al., 2018; Chen et al., 2021; Zhang and Choi, 2021; Liška et al., 2022) challenged this assumption and proposed QA datasets that specify the temporal context (e.g., *who was the President of the U.S. in 1940?*). We extend these recent efforts on time-sensitive QA to fulfill real-time, more instantaneous information needs from users: we establish

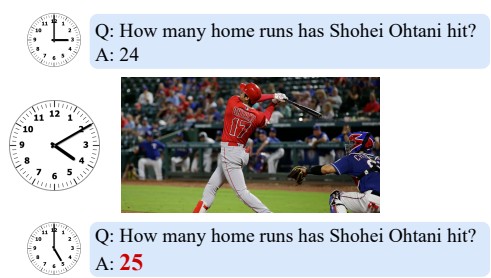

Figure 1: REALTIME QA establishes a framework to benchmark question answering at the present time: answers (e.g., the number of Shohei Ohtani's home runs) change in real time. Source: `https://thecomeback.com/mlb/shohei-ohtani-home-runs-tommy-john.html`.

---

* Work was done during JK's internship at AI2.

[2] `https://realtimeqa.github.io/`.

37th Conference on Neural Information Processing Systems (NeurIPS 2023) Track on Datasets and Benchmarks.

a dynamic benchmark based on newly-published news articles—REALTIME QA—and provide a regularly-updated (weekly in the current version) evaluation platform for the research community.

We develop an annotation framework (§2) and a benchmarking timeline for real-time QA system submissions. Every week, REALTIME QA retrieves news articles and human-written, multiple-choice questions from news websites (CNN, THE WEEK, and USA Today), covering a wide range of topics, including politics, business, sports, and entertainment. We upload these data, as well as our baseline results, to our website, and any model submission can be evaluated until the next set of questions is posted. This dynamic scheme contrasts with the well-established QA annotations (Chen et al., 2017; Chen and Yih, 2020) that are performed only *once* with information available at the time. Such annotations are effective for factoid (Berant et al., 2013; Hermann et al., 2015; Rajpurkar et al., 2016; Joshi et al., 2017) or commonsense questions (Zellers et al., 2018, 2019; Talmor et al., 2019; Sakaguchi et al., 2020), but not the real-time information needs that are our target.

We present two classes of real-time baseline systems that are built on strong, recent models (GPT-3: Brown et al., 2020; T5: Raffel et al., 2020; BART: Lewis et al., 2020a): open-book and closed-book QA models. We present a prompting method to use GPT-3 for open-domain QA. The former class uses an external knowledge source, such as Wikipedia (Min et al., 2019; Guu et al., 2020; Lewis et al., 2020b; Izacard and Grave, 2021) or news articles. The latter class of

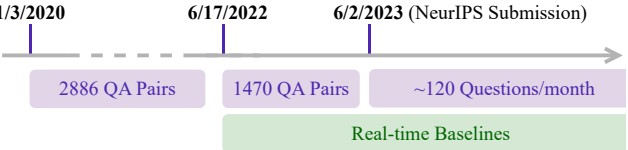

Figure 2: REALTIME QA data statistics as of June 2, 2023. We started our real-time baselines on June 17, 2022 (§2.4). We also provide past 2,886 QA pairs that can be used by model developers (e.g., finetuning).

closed-book models directly outputs an answer to each question. By design, these closed-book baselines have no access to information more recent than the time of pretraining or finetuning, thereby helping us understand the degree to which real-time information is truly necessary. Notably, a small number of questions in REALTIME QA (~12%) do not strictly require recent information; for example, Shohei Ohtani hits a home run today, leading one to ask where he was born. This is consistent with information-seeking, naturally-occurring scenarios that we target in this work, as seen in Clark et al. (2020). Most users of a QA system do not exclusively ask time-sensitive questions, even though these questions may be stimulated by current events; QA systems *should* aim to address these questions as well.

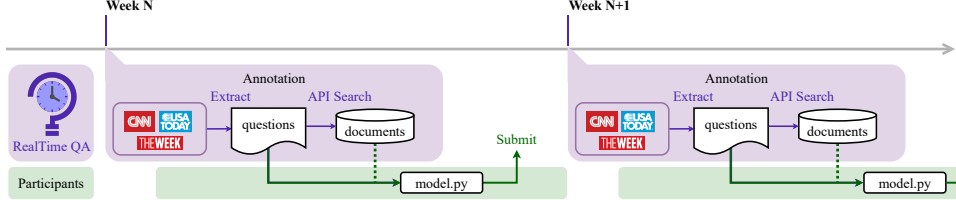

Figure 3: REALTIME QA annotation framework and submission workflow. At 3 am GMT on every Saturday, we extract questions from news websites and post them on our website. We immediately run API search for these questions (Google custom search) and share the results as a document pool. The use of this document pool is optional (indicated by a dashed line); participants are allowed to retrieve evidence documents by themselves. All evaluations are done on our website, and the submission window closes when the next set of questions is announced.

We evaluate six baselines both in multiple-choice and generation settings *in real time* and report the results over the period of June 17 through June 2, 2023. These evaluation data resulted in a total of 1,470 QA pairs (Fig. 2). Further, we provide 2,886 QA pairs that are collected in the same way but preceded our real-time evaluations. These can be used in later work for model development (e.g., finetuning). Our results show that an open-book GPT-3 model augmented with up-to-date text retrieval substantially outperforms closed-book baselines, as well as open-book models with retrieval from a past Wikipedia dump (Lewis et al., 2020b). This result illustrates that large language models

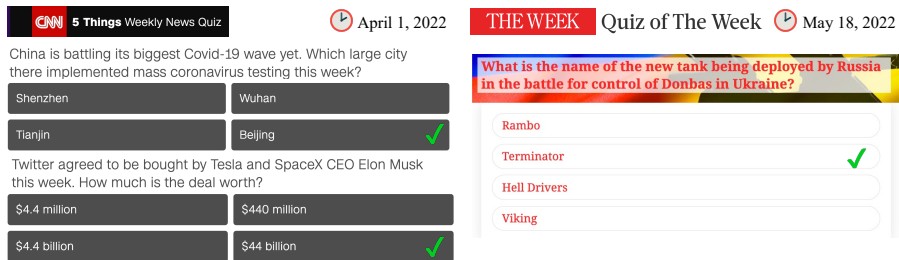

Figure 4: Examples of weekly quizzes from CNN and THE WEEK that are extracted during annotations of REALTIME QA. They span diverse genres, including politics, business, and entertainment.

can adjust their knowledge, based on the retrieved passages (§3). Nonetheless, we find that they still struggle, especially when the multiple choices include uncertainty (e.g., "none of the above"). Most of the errors originate from retrieval, rather than reading comprehension. The REALTIME QA benchmark, therefore, highlights the importance of fast, up-to-date text retrieval (Seo et al., 2019) to better serve instantaneous information needs. We share all data and code to reproduce our baselines so that follow-up work can build upon our first attempts to tackle this task.

REALTIME QA can also serve as an important step toward much-needed, broader, real-time applications of NLP. For example, a QA system with timely updates can improve emergency management of natural disasters (Imran et al., 2013, 2015, 2016; Nguyen et al., 2016) or pandemics (e.g., COVID-19; Wang et al., 2020; Lee et al., 2020; Möller et al., 2020; Alzubi et al., 2021). With the advent of online news, prior work developed automated systems that regularly retrieve and summarize news articles from the Internet (Allan et al., 2001; Radev et al., 2001; McKeown et al., 2002, 2003; Evans et al., 2004). Models developed for the REALTIME QA task can be further enhanced with such retrieval/summarization systems. We hope that our REALTIME QA interface and baseline models will serve as a useful platform for research and real-world applications.

## 2 REALTIME QA Framework

Our current version announces questions every week, based on news articles published within the past week. Here we establish the workflow (§2.1) and the framework for annotations (§2.2) and evaluations (§2.3). We then discuss our built-in baselines (§2.4) that are continually evaluated every week. Our user interface and more detailed statistics (e.g., genres and answer types) are available in Appendices B and C.

### 2.1 Workflow

Fig. 3 depicts the REALTIME QA workflow for each week. We announce ~30 multiple-choice questions at 3 am GMT every Saturday. We internally run API search (Google custom search, GCS) for these questions and share a set of documents (mostly news articles) with the URLs that are available at that time. Participants run their model on these questions, optionally using the documents from our API search as a knowledge source (indicated as dashed lines in Fig. 3). While we provide our document set to lower barriers to submission, **participants are also allowed to create and use knowledge sources by themselves** (e.g., custom retrieval models or other external APIs such as Twitter API). System submissions are shared on our website with their performance and submission time. The submission window closes when the new set of questions is announced the next week.

Note that fair, *retroactive* comparisons of systems are also possible, as long as they use data available when the submission window was still open. For instance, participants might be interested in evaluating their model against a past submission on the Week N questions. In this case, they can do so by ensuring that their system only relies on data up to Week N and simulating how their system *would have performed* at that time. Our platform still focuses on real-time evaluations and encourages every participant to submit real-time results to better reflect real-world applications.

## 2.2 Annotation

**Question Extraction** The authors of this paper perform weekly annotations in a human-in-the-loop way. We first find web pages for "weekly quizzes" from three news websites: CNN (US-based), USA Today, and The WEEK (UK-based).[3] Shown in Fig. 4 are examples that span politics and business genres. We then execute our extraction script to collect multiple-choice questions. Each of these three websites posts ~10 questions per week, resulting in ~120 questions in total every month. Weekly quizzes are also available from the New York Times and ABC Australia, but they are not included in the current version, due to issues with automatic extraction or a paid subscription system.

**API Search** Using each of these questions as a retrieval query, we run Google custom search[4] to collect the top-10 documents from the web. The retrieval target is all articles from CNN, USA Today, and THE WEEK. We then parse every document using the `newspaper3k` package[5] and store the text as well as metadata, such as the publication date and author name. In some rare cases, articles from the search get taken down, in which case we disregard them. This indeed illustrates a unique challenge of real-time applications with constantly-changing, dynamic information.

## 2.3 Evaluation

**Multiple Choice** Since REALTIME QA is a multiple-choice question dataset, we can simply measure performance by accuracy. We also explored a NOTA (none of the above) setting: one of the original choices is randomly replaced with "none of the above," thereby helping prevent models from exploiting heuristics (Rajpurkar et al., 2018). As expected, the NOTA setting resulted in performance degradation across the board (§3). NOTA choices can be found in other multiple-choice QA or reading comprehension datasets (Richardson et al., 2013; Lai et al., 2017).

**Generation** We also experiment with a generation setting where no choices are given, to better reflect real-world applications. Under this setting, we evaluate performance with exact matching and token-based F1 scores, following the standard practice in question answering (Rajpurkar et al., 2016).

**Human Performance** We randomly sampled 10 weeks from June 17, 2022 through January 13, 2023 (300 questions in total), and the authors of this paper answered multiple-choice questions using Google search. This resulted in the accuracy of $96.7\%$. Most questions in REALTIME QA are straightforward (e.g., single-hop questions) and a human with Internet access can easily answer them.[6] For the sustainability of the dynamic benchmark, we do not provide an estimate of human performance on a regular basis.

## 2.4 Real-time Baselines

REALTIME QA executes six baselines in real time that are based on strong pretrained models: four open-book and two closed-book models. These six models are evaluated and made publicly available when weekly questions are announced. Any submission to REALTIME QA is compared against them. Participants can also build their model upon our baselines. See Appendix §A for more detail.

### 2.4.1 Open-book QA Models

Open-book QA models follow a two-step pipeline: **document retrieval** that finds evidence documents from an external knowledge source (e.g., Wikipedia) and **answer prediction** (or reading comprehension) that outputs an answer conditioned on the question and evidence documents. For either step, we experiment with two variants, resulting in a total of four configurations. Open-book systems have the advantage of being capable of updating the external knowledge source at test time (Lewis et al., 2020b). This property is particularly crucial for questions in REALTIME QA that inquire about information at the present time.

---

[3]Fair use is allowed under Section 107 of the Copyright Act in the U.S.: `https://www.copyright.gov/title17/92chap1.html#107`.

[4]`https://programmablesearchengine.google.com/`.

[5]`https://github.com/codelucas/newspaper`.

[6]In fact, USA Today has a record of human top scorers every week, and they all get perfect scores. E.g., `https://www.usatoday.com/storytelling/quiz/news-quiz/2022-07-01/`.

**Document Retrieval** For the retrieval step, we experiment with two configurations: top-5 Wikipedia documents from dense passage retrieval (**DPR**; Karpukhin et al., 2020) and top-5 news articles from **GCS** (§2.2). In DPR, English Wikipedia articles from the December 2018 dump are segmented into 100-word documents (Wang et al., 2019). DPR encodes the question and every document into 768-dimensional vectors; it then computes the inner product to obtain a matching score and selects documents with top-5 matching scores. We use the BERT-based model (Devlin et al., 2019), finetuned on the Natural Questions dataset (Kwiatkowski et al., 2019) from the Hugging Face Transformers library (Wolf et al., 2020). GCS uses an external API, and we found that it sometimes returned fewer than five documents (~10% of the time); in this case, we add top documents from DPR to create a top-5 document set.

**Answer Prediction** We explore two methods for answer prediction, conditioned on the question and the corresponding retrieved text: retrieval-augmented generation (**RAG**; Lewis et al., 2020b) and a prompting method with **GPT-3** (text-davinci-002; Brown et al., 2020). In the multiple-choice setting, we compute the log probability of every choice and normalize it by the generation sequence length. We then select the choice with the best score. For the generation setting, we simply perform text decoding.

For the **RAG** baseline, we use the BART-based (Lewis et al., 2020a) RAG-sequence model, again finetuned on Natural Questions from the Transformers library. This model predicts the answer sequence $\mathbf{y}$ autoregressively from left to right while marginalizing over the set of top-5 retrieved documents ($\mathcal{Z}$): $P(\mathbf{y}) = \sum_{z \in \mathcal{Z}} P(z) \prod_{t=1}^{|\mathbf{y}|} P(y_t | z, \mathbf{y}_{\leq t})$. Here $P(z)$ is given by the matching score from the retrieval step.[7] In the equation, the conditioned-upon question is suppressed for brevity.

We propose a straightforward **GPT-3** prompting method with temporal contexts (Fig. 5).[8] We prepend to every question the title and the first two paragraphs of the top-5 articles from the document retrieval step.[9] The publication date is inserted, using the metadata of each retrieved article (e.g., "Article on November 2, 2021" in Fig. 5). For Wikipedia passages retrieved by DPR, we prepend "December 31, 2018," based on the Wikipedia dump date (Karpukhin et al., 2020). Our ablation studies on date insertion will show that the open-book GPT-3 system benefits from specifying the dates of the question and the retrieved articles to some extent (§3.2).

### 2.4.2 Closed-book QA Models

Closed-book QA models directly answer questions without access to external knowledge. They have proven competitive with open-book models on some QA datasets (Roberts et al., 2020; Guu et al., 2020). Since these models are trained/finetuned on the data available at that time, they cannot address questions about new events or updated information. Nonetheless, some of the real-time information needs do not necessarily require up-to-date information. Indeed, REALTIME QA contains a small portion of such questions (~10%). For in-

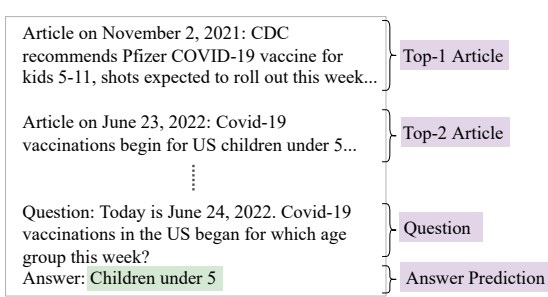

Figure 5: Example prompt for answer generation with the open-book GPT-3 baseline. For the closed-book GPT-3 baseline, the top-5 articles are not given. We perform ablation studies on the date information (§3.2).

stance, *Microsoft retired its Internet Explorer browser this week. What year did it debut?* Such questions are triggered by a new event but inquire about facts in the past that have not changed recently. Most users of a QA system do not exclusively raise time-sensitive questions, and QA systems should aim to address these questions as well. Closed-book baselines thus quantify the degree to which up-to-date information is necessary to answer questions in REALTIME QA. We use the following two strong methods for closed-book QA.

---

[7]Unlike DPR, GCS does not provide matching scores. We treat top-5 documents with equal probabilities.

[8]See Lazaridou et al. (2022) for other prompt templates.

[9]This substantially reduces the inference cost. They contain most of the key information in each article.

**Finetuning Method**  We use the T5 model (T5-11B; Raffel et al., 2020) finetuned on the Natural Questions data, again from the Transformers library. Following the open-book baseline, we select the choice with the maximum average log score in the multiple-choice setting.

**Prompting Method**  Similar to the open-book baselines (§2.4.1), we apply a prompting method to GPT-3 (text-davinci-002). We use the same prompt as Fig. 5, except that no articles are inserted before the question. Again following the open-book baselines, we select the choice with the maximum average log score in the multiple-choice setting.

## 3  Experiments and Analysis

We started our real-time experiments on June 17 2022, spanning a year as of June 2 2023 (1470 questions in total). We will continue our weekly annotations, but here we report our experimental and analysis results so far and give guidance to future participants.

Table 1: Results from the past year (from June 17, 2022 through June 2, 2023). GCS: Google custom search; DPR: dense passage retrieval (Karpukhin et al., 2020); RAG: retrieval-augmented generation (Lewis et al., 2020b).

| Real-time Baselines | | Multi-choice | | Generation | |
|---|---|---|---|---|---|
| Retrieve | Predict | Orig. | NOTA | EM | F1 |
| Open DPR | RAG | 27.4 | 24.8 | 2.4 | 4.1 |
| DPR | GPT-3 | 43.9 | 35.8 | 13.3 | 19.7 |
| GCS | RAG | 46.9 | 37.9 | 17.5 | 22.1 |
| GCS | GPT-3 | **66.5** | **58.4** | **34.6** | **45.3** |
| Closed — | T5 | 39.1 | 35.3 | 9.7 | 14.7 |
| — | GPT-3 | 44.9 | 34.1 | 15.3 | 22.3 |

Table 2: Ablation studies on date insertion in the prompt for the open-book (Google custom search; GCS) and close-book GPT-3 baselines. All results are averaged over the first six weeks: June 17 through July 22, 2022.

| Date Insert | | Multi-choice | | Generation | |
|---|---|---|---|---|---|
| Articles | Qs | Orig. | NOTA | EM | F1 |
| Open ✓ | ✓ | **69.3** | 59.8 | **28.7** | **39.4** |
| ✓ | ✗ | 66.5 | **62.6** | 24.7 | 36.3 |
| ✗ | ✓ | 67.0 | 57.5 | 28.1 | 38.2 |
| ✗ | ✗ | 65.9 | 61.5 | **28.7** | 38.3 |
| Closed — | ✓ | 39.7 | 31.3 | 7.3 | 15.2 |
| — | ✗ | 45.8 | 38.5 | 9.0 | 15.9 |

### 3.1  Results

Seen in Table 1 are the results from the past year. In all three settings (original/NOTA multiple choice and generation), GPT-3 with Google custom search (GCS) retrieval achieves the best performance. In particular, GPT-3 with GCS substantially outperforms both closed-book GPT-3 and GPT-3 with DPR (from a December 2018 Wikipedia dump): e.g., 34.6 vs. 15.3/13.3 in generation exact matching. This suggests that GPT-3 is able to answer questions based on the given prompt, rather than relying on past information from pretraining. Nevertheless, we still see a large performance drop of all baselines from the original multiple-choice setting to NOTA ("none of the above"): e.g., 58.4 vs. 66.5 for GPT-3 with GCS retrieval. Future work can further improve GPT-3's ability of reading comprehension, especially regarding answer uncertainty.

### 3.2  Analysis and Ablations

**Date Insertion for Prompting**  Our prompt for the GPT-3 baselines prepends date information both to the articles and question (Fig. 5). Table 2 shows results from ablation studies on date insertion for the open-book (GPT-3 with Google custom search) and closed-book GPT-3 models. Temporal specification almost always helps the open-book GPT-3 model. Interestingly, it hurts the performance of the closed-book model, perhaps because the specified date is generally unseen during pretraining and the prompt becomes "out-of-domain."

**Error Breakdown**  We conducted a manual error analysis of the results so far. In particular, we categorized answers from the best generation model (open-book GPT-3 with GCS) into three categories: correct, retrieval error, and reading comprehension error. For the questions from the first six weeks, the breakdown was the following: correct (52%), retrieval error (34%), and reading comprehension error (13%). This suggests that the key to instantaneous applications of question answering is **accurate, up-to-date information retrieval**.

**Performance vs. Submission Time** Fig. 6 plots the performance of the open-book GPT-3 baseline with Google custom search (GCS) over varying submission (i.e., GCS retrieval) time. All results are averaged over the questions between June 17 and July 22, 2022. We see a consistent pattern: the performance remains high (or improves) up to around 24 hours after the announcement but substantially degrades later. While the performance can improve when GCS starts to retrieve more recent articles, it eventually suffers from temporal gaps. Our website provides the submission time of every system as well as its performance.

**Examples** Table 3 shows some examples that compare the closed-book and open-book GPT-3 models. The first three examples illustrate that GPT-3 can correctly update its answer based on the retrieved documents across diverse genres: natural disasters, the COVID-19 pandemic, and entertainment. The last three cases, on the other hand, demonstrate a critical limitation of current large language models in temporal understanding: **the retrieved documents do not suffice to answer the questions due to a temporal gap, and GPT-3 still generates an outdated answer**. Ideally, GPT-3 should inform the user or even the retrieval module that it does not have enough evidence to answer the question. This way, the retrieval module can expand its search, or the user can consult other resources.

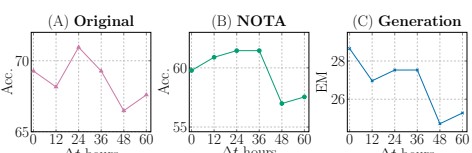

Figure 6: Performance vs. submission time (hours after the announcement of questions, 3 am GMT on Saturday) over the three evaluation settings (A: original multiple choice; B: none of the above; C: generation). All results are from open-book GPT-3 with Google custom search (GCS) and averaged over the questions from June 17, 2022 through July 22, 2022. $\Delta t = 0$ for all of our six real-time baselines by default.

Note that it is possible to limit the retrieval target to recent articles,[10] but there are potential failure modes. Firstly, some questions in REALTIME QA inquire about the past, and models can benefit from older articles when answering such questions. Further, the appropriate date range for retrieval varies from question to question in real-world applications; some questions inquire about this year, while others about this week. We thus do not implement such filtering for the current real-time baselines.

# 4   Related Work

REALTIME QA has time sensitivity, which several prior works addressed on various NLP tasks. Here we discuss its relation to long-standing summarization and text retrieval tasks, as well as recent work on temporal misalignment between training and evaluation. We then discuss its connections to dynamic evaluations and open-domain QA.

**Summarization/Retrieval over Time** Temporal (or timeline) summarization is a task that retrieves documents from the web and provides their summary *over time* (Catizone et al., 2006; Aslam et al., 2013, 2014, 2015; Martschat and Markert, 2017, 2018). Update summarization (Witte et al., 2007; Dang and Owczarzak, 2008) and new event detection/track (Allan et al., 1998; Li et al., 2005) are tasks that monitor and track newly-added information. Prior work created datasets and systems for these tasks (Tran et al., 2013, 2015; Wang et al., 2015; Chen et al., 2019; Gholipour Ghalandari and Ifrim, 2020). Their evaluations are usually executed *statically*, with information available at the time of data collection.

In contrast, the TREC real-time summarization track evaluates systems in real time during a 1–2 week evaluation period (Lin et al., 2016, 2017; Sequiera et al., 2018). Several other works and initiatives focused particularly on financial news summarization (Filippova et al., 2009; Passali et al., 2021) or emergency management technology (Temnikova et al., 2014; Ghosh et al., 2017; McCreadie et al., 2019), including the COVID-19 pandemic (Buntain et al., 2020; Pasquali et al., 2021). This work regularly evaluates question answering systems over diverse topics, but we share the goal of dealing with novel and evolving information over time; retrieval or summarization methods from these tasks (e.g., Yan et al., 2011a,b, 2012; Shou et al., 2013) can be combined with models in REALTIME QA to serve various time-sensitive information needs from users. REALTIME QA can also be used to evaluate time-sensitive retrieval systems by the downstream QA performance.

---

[10]Indeed, GCS has a *paid* version with a date range feature that filters retrieval results by date.

Table 3: Examples that compare closed-book and open-book GPT-3 answers with top-5 articles from Google custom search (GCS) retrieval. As in the first three examples, GPT-3 can adjust its answer based on newly-retrieved documents. When the retrieved documents are *outdated* or unrelated, however, GPT-3 ignores the temporal gap and yields an outdated answer.

| Question | Retrieved Documents (Top-5) | |
| --- | --- | --- |
| Historic rainfall led to flooding, mudslides and visitor evacuations at which national park?
**Date**: June 17, 2022
**Answer**: Yellowstone National Park
**Closed GPT-3**: Yosemite National Park
**Open GPT-3**: Yellowstone National Park | **June 14, 2022**
Yellowstone National Park flooding 'still raging'...
**June 13, 2022**
Yellowstone National Park closes entrances, evacuates visitors amid 'unprecedented' rainfall...
**June 15, 2022**
Dozens evacuated as unprecedented flooding forces Yellowstone National Park to close... | **June 15, 2022**
Yellowstone still closed as flooding recedes and thousands evacuate...
**June 14, 2022**
Home swept away as Yellowstone National Park is hit by major floods and mudslides... |
| Covid-19 vaccinations in the US began for which age group this week?
**Date**: June 24, 2022
**Answer**: Children under 5
**Closed GPT-3**: 18 and up
**Open GPT-3**: Children under 5 | **November 2, 2021**
CDC recommends Pfizer COVID-19 vaccine for kids 5-11, shots expected to roll out this week...
**June 23, 2022**
Covid-19 vaccinations begin for US children under 5... | **July 22, 2021**
Biden says kids under 12 could be eligible for COVID vaccines in weeks...
**November 10, 2021**
COVID-19 cases on the rise again in Iowa...
**November 1, 2021**
Everything to know about COVID-19 vaccine and children... |
| Which wildly popular show was recently green lit for a new season?
**Date**: June 17, 2022
**Answer**: Squid Game
**Closed GPT-3**: The show "Game of Thrones" was recently green lit for a new
**Open GPT-3**: Squid Game | **June 12, 2022**
Netflix green lights 'Squid Game' season 2...
**June 17, 2022**
5 things to know for June 17...
**June 4, 2019**
'Looking for Alaska' details revealed for Hulu limited series... | **February 4, 2022**
The Busch Light Clash goes green this weekend...
**September 26, 2018**
Dip into 4 new mysteries for fall, including Kate Atkinson's spy novel 'Transcription'... |
| The IRS announced it will do what this week?
**Date**: June 24, 2022
**Answer**: Finish processing the backlog of 2021 tax returns
**Closed GPT-3**: The IRS announced it will begin processing tax returns this week.
**Open GPT-3**: The IRS announced it will begin processing 2021 tax returns as soon as Jan. 24 | **January 10, 2022**
IRS 2022 tax season set to begin 2 weeks early on Jan. 24...
**March 12, 2021**
When will I get my third stimulus check?...
**March 22, 2021**
IRS says more stimulus checks on the way... | **February 11, 2022**
Don't panic if you got a scary IRS notice...
**January 10, 2022**
IRS will begin processing 2021 tax returns as soon as Jan. 24 |
| Which country is now "bankrupt," according to a statement this week from its administration?
**Date**: July 8, 2022
**Answer**: Sri Lanka
**Closed GPT-3**: Greece
**Open GPT-3**: Venezuela | **March 2, 2022**
Gun manufacturers are not entirely exempt from being sued... the now-bankrupt gun manufacturer...
**March 12, 2021**
Mitch McConnell seeks to end Democrat's 'crazy policy' of beefed-up unemployment benefits... let states go bankrupt... | **March 20, 2022**
Half of US hotels could close amid coronavirus crisis... hotels around the country go bankrupt...
**September 26, 2013**
Colo. farmers arrested... the now-bankrupt Jensen Farms...
**January 10, 2022**
Trump administration restrictions on asylum are cruel... Immigration policy is morally bankrupt... |
| Which head of state announced his resignation this week?
**Date**: July 8, 2022
**Answer**: UK Prime Minister Boris Johnson
**Closed GPT-3**: Japanese Prime Minister Shinzo Abe announced his resignation this week.
**Open GPT-3**: Andrew Cuomo | **August 11, 2021**
NY Gov. Andrew Cuomo will resign in two weeks...
**September 21, 2021**
Maricopa County Supervisor Steve Chucri to resign...
**January 25, 2016**
Ball State president Ferguson resigns... | **March 23, 2021**
Oregon State University President F. King Alexander resigns...
**August 10, 2021**
NY Gov. Andrew Cuomo to resign amid scandal... |

**Temporal Misalignment and Degradation** While not particularly motivated by instantaneous information needs like REALTIME QA, prior work also explored temporal aspects of a variety of NLP tasks. A flurry of recent work analyzed performance degradation from temporal misalignment between (pre)training and evaluation/deployment on many NLP tasks (Lazaridou et al., 2021; Röttger and Pierrehumbert, 2021; Luu et al., 2022; Onoe et al., 2022) and proposed mitigation methods (Huang and Paul, 2018, 2019; Dhingra et al., 2022; Jang et al., 2022a,b; Lee et al., 2022). An open-book QA model conditions answer generation upon newly-retrieved documents (Lewis et al., 2020b), but the extent to which answer generation can be updated based on the retrieved documents is limited (Longpre et al., 2021b). Temporal degradation is, therefore, one of the challenges that models in REALTIME QA need to address.

**Dynamic Benchmarks** Unlike the majority of datasets in natural language processing, REALTIME QA evaluates systems *dynamically* and its evaluations change over time. Several other prior works update challenge test sets (Kiela et al., 2021; Potts et al., 2021; Ma et al., 2021), evaluation tasks (Thrush et al., 2022), or metrics (Gehrmann et al., 2021, 2022; Mishra and Arunkumar, 2021; Kasai et al., 2022). REALTIME QA hosts a similar online platform and adopts a dynamic scheme specifically to pursue instantaneous applications.

**Open-Domain QA** Much prior work proposed datasets for open-domain QA for English and beyond (Clark et al., 2020; Asai et al., 2021, 2022; Longpre et al., 2021a; Zhang et al., 2021). Several recent works challenged the conventional problem setups (Chen and Yih, 2020) where correct answers can be found from a fixed, external knowledge source, such as Wikipedia. Similar to REALTIME QA, Zhang and Choi (2021); Liška et al. (2022) focused on temporal or geographical contexts that can change the answer to the same question. Consistent with these prior efforts, REALTIME QA aims toward broader applications of question answering beyond the conventional framework.

# 5 Conclusion and Future Work

We introduce REALTIME QA, a dynamic, open-domain QA benchmark that asks questions at the present time. Our platform announces questions every week and continually evaluates six real-time baselines. Our experiments from the first year suggest that accurate, up-to-date information retrieval is particularly important to serve speedy information needs. We hope that REALTIME QA encourages research efforts toward fast, accurate applications of natural language processing.

# Limitations

This work aims to develop a QA benchmark for addressing instantaneous information needs, including emergency management. The current version of REALTIME QA has two major limitations due to our annotation framework (§2.2): 1) question/answer pairs are all written in English, and the covered topics tend to be English-centric (US and UK); 2) questions are announced on a weekly basis, rather than a truly instantaneous basis. Nevertheless, our benchmark departs from many static datasets from prior work and provides an important step towards the research goal. We hope to develop future versions of REALTIME QA that mitigate these limitations.

# Acknowledgements

We thank Noriyuki Kojima, Alisa Liu, Ofir Press, Koji Shiono, Wenya Wang, the ARK group at the UW, and the Mosaic team at the Allen Institute for AI for their helpful feedback on this work.

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

# Appendices

## A  Baseline Configurations

We provide the configurations for our real-time baselines (§2.4). We use the open-source, Transformers library and ensure easy replication of our results. Table 4 lists the configurations for dense passage retrieval (Karpukhin et al., 2020) and retrieval-augmented generation (Lewis et al., 2020b). We generally follow the default settings from the Transformers library. Seen in Table 5 is the configuration for the closed-book T5 baseline. We again generally follow the default setting.

Table 4: Configurations for dense passage retrieval (Karpukhin et al., 2020) and retrieval-augmented generation (Lewis et al., 2020b) from the Transformers library (Wolf et al., 2020).

| Option | Value |
|---|---|
| n_docs | 5 |
| max_combined_length | 300 |
| retrieval_vector_size | 768 |
| retrieval_batch_size | 8 |
| is_encoder_decoder | True |
| prefix | None |
| bos_token_id | None |
| pad_token_id | None |
| eos_token_id | None |
| decoder_start_token_id | None |
| title_sep | '/' |
| doc_sep | '//' |
| dataset | 'wiki_dpr' |
| dataset_split | 'train' |
| index_name | 'compressed' |
| index_path | None |
| passages_path | None |
| use_dummy_dataset | False |
| reduce_loss | False |
| label_smoothing | 0.0 |
| do_deduplication | True |
| exclude_bos_score | False |
| do_marginalize | False |
| output_retrieved | False |
| use_cache | True |
| forced_eos_token_id | None |

## B  REALTIME QA Interface

Fig. 7 shows our REALTIME QA interface. It gets updated every week, and all six baselines are evaluated as soon as the questions are available. Submissions will be shown on the same page, together with their submission time.

## C  REALTIME QA Statistics

Table 6 provide more detailed statistics of REALTIME QA from the first four weeks. We analyze the questions from the first four weeks along genres and answer types. We also found that ~10% of the questions were not strictly time-sensitive. These questions include, for example, "Temperatures in Britain are set to soar this weekend, but what is the hottest UK temperature on record?" from June 17, 2022. We do not filter out these cases to simulate information-seeking, naturally-occurring scenarios.

Table 5: Configuration for the closed-book T5 baseline (Raffel et al., 2020) from the Transformer library.

| Option | Value |
|---|---|
| _name_or_path | /home/patrick/t5/t5-11b-ssm-nq |
| architectures | [ "T5ForConditionalGeneration"] |
| d_f | 65536 |
| d_kv | 128 |
| d_model | 1024 |
| decoder_start_token_id | 0 |
| dropout_rate | 0.1 |
| eos_token_id | 1 |
| feed_forward_proj" | relu |
| initializer_factor | 1.0 |
| is_encoder_decoder | True |
| layer_norm_epsilon | 1e-06 |
| model_type | t5 |
| num_decoder_layers | 24 |
| num_heads | 128 |
| num_layers | 24 |
| output_past | True |
| pad_token_id | 0 |
| relative_attention_num_buckets | 32 |
| tokenizer_class | T5Tokenizer |
| vocab_size | 32128 |

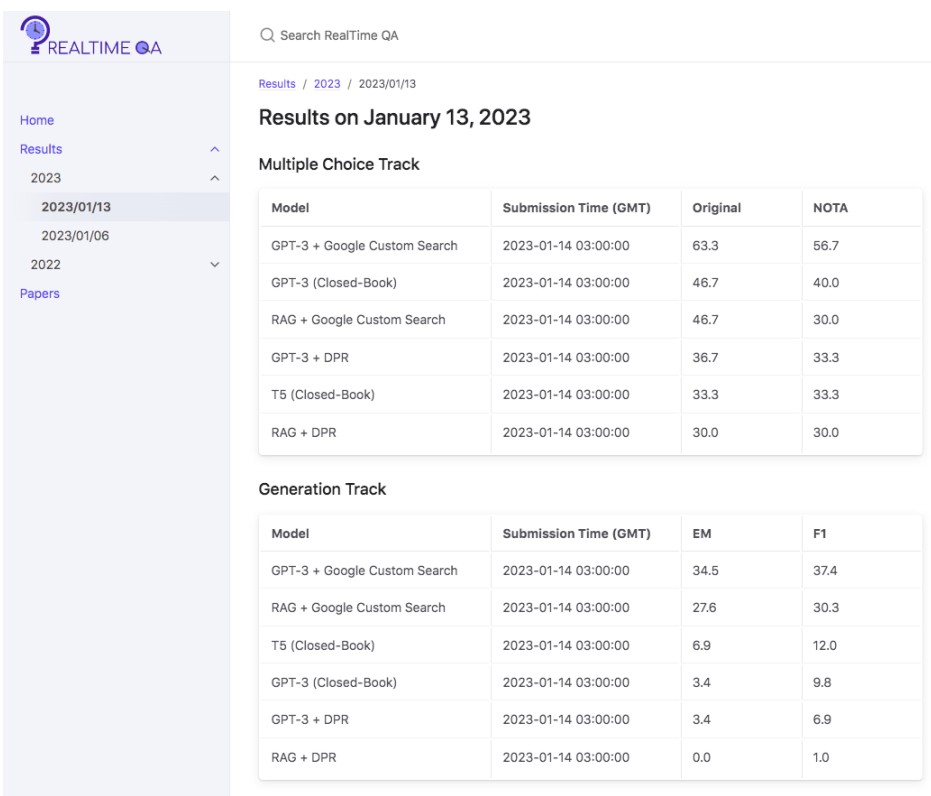

Figure 7: REALTIME QA interface. It is updated every week, and all six baselines are evaluated as soon as the questions are available. Submissions will be shown on the same page, together with their submission time.

Table 6: Detailed statistics (%) of REALTIME QA. We analyze the questions from the first four weeks along genres and answer types. We also found that ~12% of the questions were not strictly time-sensitive. These questions include, for example, "Temperatures in Britain are set to soar this weekend, but what is the hottest UK temperature on record?" from June 17, 2022. We do not filter out these cases to simulate information-seeking, naturally-occurring scenarios.

| **Genre** | | | | | | |
|---|---|---|---|---|---|---|
| **Politics** | **Business** | **Entertain** | **Science** | **Technology** | **Health** | **Disaster** |
| 36.9% | 17.5% | 17.5% | 7.0% | 7.0% | 8.8% | 5.2% |
| **Answer Type** | | | | | | |
| **Person** | **Location** | **Time** | **Number** | **Organization** | **Event** | **Miscellaneous** |
| 12.3% | 19.2% | 5.3% | 22.8% | 3.5% | 8.8% | 28.1% |

