# REALTIME QA Datasheet

**Jungo Kasai**[♡♣]    **Keisuke Sakaguchi**[◇♣]    **Yoichi Takahashi**[◇]    **Ronan Le Bras**[♣]    **Akari Asai**[♡]
**Xinyan Velocity Yu**[♡]    **Dragomir Radev**[♥]    **Noah A. Smith**[♡♣]    **Yejin Choi**[♡♣]    **Kentaro Inui**[◇♣]

[♡]Paul G. Allen School of CSE, University of Washington  [♣]Allen Institute for AI
[◇]Tohoku University    [♥]Department of Computer Science, Yale University    [♠]RIKEN

REALTIME QA

realtimeqa.nlp@gmail.com    @realtimeqa

## 1 Motivation for Datasheet Creation

**Why was the dataset created?**    The REALTIME QA dataset was created to challenge conventional assumptions in open-domain Question Answering (QA) datasets and pursue real-time, instantaneous applications. Most existing QA datasets are static and do not account for the continuously evolving nature of world events and information. Therefore, REALTIME QA was created to provide a dynamic platform that asks questions about the current world, challenging QA systems to provide answers about novel events and fresh information. This is an attempt to better emulate the real-time needs of practical QA systems, which require up-to-date information to provide relevant and accurate responses. Furthermore, the results from REALTIME QA may identify areas of potential research, such as improving how QA systems deal with unanswerable cases based on current information retrieval.

**Has the dataset been used already?**    The REALTIME QA dataset has already been used. The authors employed the dataset to build strong baseline models using large pretrained language models, including GPT-3 and T5. The dataset was used in real-time evaluations over the past year. The results of these experiments highlight the importance of up-to-date information retrieval and indicate areas where the QA systems, specifically GPT-3 in this context, could be improved. For instance, it was noted that GPT-3 tends to return outdated answers when the retrieved documents don't provide sufficient information, highlighting a potential area for future research.

**Who funded the dataset?**    REALTIME QA was funded by the Masason fellowship to Jungo Kasai.

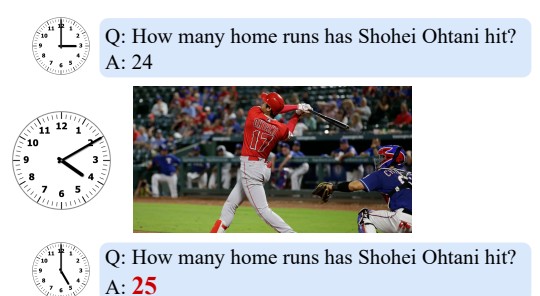

Figure 1: REALTIME QA establishes a framework to benchmark question answering at the present time: answers (e.g., the number of Shohei Ohtani's home runs) change in real time. Source: https://thecomeback.com/mlb/shohei-ohtani-home-runs-tommy-john.html.

## 2 Dataset Composition

**What are the instances?**    Given that it is a dynamic and real-time dataset, an instance is a newly announced question about current events or new information, along with the system's answer to that question. This could also include the system's interactions with its information retrieval module (for instance, the documents it retrieves to help answer the question) and any information about whether the answer was correct or not, or whether the system determined the question to be unanswerable based on the current information available to it.

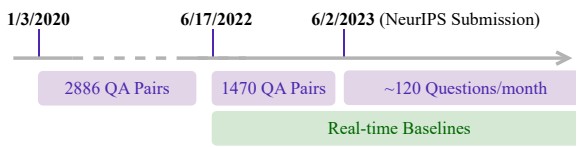

Figure 2: REALTIME QA data statistics as of January 13, 2023. We started our real-time baselines on June 17, 2022 (§**??**). We also provide past 2,886 QA pairs that can be used by model developers (e.g., finetuning).

**How many instances are there?** Shown in Table 2 are data statics. We have 1470 question-answer pairs as of now. We have about 120 new question-answer paiers per month.

**What data does each instance consist of?** Considering its dynamic nature and focus on real-time information, an instance includes:

- A question about a current event or new piece of information.

- The answer generated by the QA system, such as GPT-3 or T5.

- Any associated metadata, like the time of the question generation and answer production.

**Does the data rely on external resources?** Data are created based on news websites (CNN, THE WEEK, and USA Today). See our paper for more detail.

**Are there recommended data splits or evaluation measures?** We focus on evaluations, so most data should be used for evaluation purposes. Nonetheless, we provide past data pontentially for finetuning.

## 3 Data Collection Process

**How was the data collected? Question Extraction** The authors of this paper perform weekly annotations in a human-in-the-loop way. We first find web pages for "weekly quizzes" from three news websites: CNN (US-based), USA Today, and The WEEK (UK-based).[1] We then execute our extraction script to collect multiple-choice questions. Each of these three websites posts ~10 questions per week, resulting in ~30 weekly questions in total. Weekly quizzes are also available from the New York Times and ABC Australia, but they are not included in the current version, due to issues with automatic extraction or a paid subscription system. **API Search** Using each of these ~30 questions as a retrieval query, we run Google custom search[2] to collect the top-10 documents from the web. The retrieval target is all articles from CNN, USA Today, and THE WEEK. We then parse every document

using the `newspaper3k` package[3] and store the text as well as metadata, such as the publication date and author name. In some rare cases, articles from the search get taken down, in which case we disregard them. This indeed illustrates a unique challenge of real-time applications with constantly-changing, dynamic information.

**Who was involved in the collection process and what were their roles?** As mentioned above, the authors maintain the benchmark dynamically every week.

**Over what time frame was the data collected?** The benchmark is an ongoing effort, and the colleciton continues to happen every week. The collection started on June 17, 2022.

**Does the dataset contain all possible instances?** Ultimately, the goal of REALTIME QA is to build a system that can answer any question in *real time*. For this reason, our dataset is obviously a sample from the infinite space.

**If the dataset is a sample, then what is the population?** This work aims to develop a QA benchmark for addressing instantaneous information needs, including emergency management. The current version of REALTIME QA has two major limitations due to our annotation framework: 1) question/answer pairs are all written in English, and the covered topics tend to be English-centric (US and UK); 2) questions are announced on a weekly basis, rather than a truly instantaneous basis. Nevertheless, our benchmark departs from many static datasets from prior work and provides an important step towards the research goal. We hope to develop future versions of REALTIME QA that mitigate these limitations.

## 4 Data Preprocessing

**What preprocessing / cleaning was done?** The authors of this paper perform weekly annotations in a human-in-the-loop way. We first find web pages for "weekly quizzes" from three news websites: CNN (US-based), USA Today, and The WEEK (UK-based). We then execute our extraction script to collect multiple-choice questions. Each of these three websites posts ~10 questions per week, resulting in ~30 weekly questions in total. Weekly quizzes are also available from the New York Times

---

[1]Fair use is allowed under Section 107 of the Copyright Act in the U.S.: https://www.copyright.gov/title17/92chap1.html#107.

[2]https://programmablesearchengine.google.com/.

[3]https://github.com/codelucas/newspaper.

and ABC Australia, but they are not included in the current version, due to issues with automatic extraction or a paid subscription system.

**Was the raw data saved in addition to the cleaned data?**   We annotate our benchmark with retrieval results from Google Custom Search and answers from strong reading comprehension models (e.g., GPT-3).

**Does this dataset collection/preprocessing procedure achieve the initial motivation?**   Yes. The REALTIME QA dataset was successful in establishing a dynamic platform that challenges QA systems with real-time information and novel events, aligning with its stated purpose. It also enabled the identification of potential areas for future research, such as improving the ability of QA systems to handle situations where retrieved documents don't provide sufficient information to answer a question.

## 5   Dataset Distribution

**How is the dataset distributed?**   It is available for downloads at https://realtimeqa.github.io/.

**When was it released?**   The annotations and release started on June 17, 2022.

**What license (if any) is it distributed under?** REALTIME QA is distributed under the CC BY-SA 4.0 license.[4]

**Who is supporting and maintaining the dataset?** be maintained by the first three authors of the paper: Jungo Kasai, Keisuke Sakaguchi, and Yoichi Takahashi. All updates will be posted on the dataset website.

## 6   Legal and Ethical Considerations

**Were workers told what the dataset would be used for and did they consent?**   Not applicable because all human annotations are done by the authors.

**If it relates to people, could this dataset expose people to harm or legal action?**   Our dataset can include incorrect information to the extent that news websites can have wrong information about people. Nonetheless, we performed extensive quality control and answer verification to minimize the risk of harming people.

---

[4]https://creativecommons.org/licenses/by-sa/4.0/legalcode

**If it relates to people, does it unfairly advantage or disadvantage a particular social group?** One fundamental problem with the recent question answering benchmarks is that most of their questions are written for static applications. As such, models trained and developed on those datasets are likely to fail to serve people with diverse needs RE-ALTIME QA partially remedies this long-standing problem by targeting real-time applications.

## 7   Author Statement

We affirm that we have complied fully with all ethical guidelines, data rights, and privacy regulations pertaining to the collection, processing, and dissemination of data.

We assure that the data contained within the RE-ALTIME QA dataset has been collected, processed, and shared according to the highest standards of ethical research, and that all necessary permissions have been sought and granted where required.

We bear full responsibility for any potential violation of rights, ethical norms, or privacy related to the data, and we commit to promptly addressing and rectifying any issues brought to our attention.