# OpenReview forum: "RealTime QA: What's the Answer Right Now?"
_NeurIPS.cc/2023/Track/Datasets_and_Benchmarks — NeurIPS 2023 Datasets and Benchmarks Poster_

### Official Review · Reviewer_CU6H · 2023-07-18
**cool idea**

**Rating:** 9
**Confidence:** 4

**Strengths:**

S1: Evaluating NLP systems against ever-changing information is a critical requirement for knowledge intensive tasks such as fact checking or question answering.

S2: Updates are provided at a weekly frequency which balances comparison of long-term effects and short term rapidly changing information.

S3: The study is well positioned with ongoing research about model performance degradation and studying models with dynamic benchmarking.

**Additional Feedback:**

N/A

**Clarity:**

Mostly, although it took a while for me to understand whether the paper is evaluating the QA up to the June 2023 cutoff, or whether the dataset will be maintained in the future. I'm still unsure about this exactly.

**Correctness:**

Minor Errors:
L60 - the date for June 17 is not provided here. Is this from 2022 too?


**Documentation:**

The data looked OK from what i saw on the github page.  Although effort invested in documentation may help wider adoption of the dataset.

**Limitations:**

Please see W2. I think this limitation is important to consider, but I think the dataset itself has enough merits in spite of this limitation.

**Opportunities For Improvement:**

W1: From my understanding of the benchmark, the authors will be providing new updates to the benchmark dataset over time and evaluating the change in model’s performance. This is a strength. But, I have a concern about the longevity of the benchmark, how long will updates be provided for? How will researchers compare accuracy or degradation of models? Would researchers compare a specific checkpoint in time? Or evaluate the variance of dAccuracy/dt?

 W2: The generation of the questions is based on a selection of western news outlets. While this does give a wide-range of topics, I would hope to see information from more marginalised communities as part of the question answering evaluation. Furthermore, it is a concern that selective reporting of the news articles means that only a subset of popular topics (e.g. War in Ukraine, Coronavirus, entertainment) are evaluated and compared.  e.g. from Table 6, it appears that more than 1/3 of the dataset focuses on politics questions.

W3: As information is generated on a weekly cadence, is there any discrepancy in accuracy between articles from the start of the week vs the end of the week?

W4: While multiple choice or string-based answer generation are OK evaluation metrics, I think it would be exciting to see future versions of this dataset which also study and evaluate the attribution to a source.

**Relation To Prior Work:**

There is extensive work in temporal QA and reasoning. This paper adds to this body of literature well by proposing a new dataset of questions that are generated from a weekly corpus of news data.

**Summary And Contributions:**

This paper presents a question answering dataset and model evaluation benchmark. In contrast to previous works in question answering, which assume a single static snapshot that is fixed at the time of annotation, this work introduces a dataset that updates at a weekly frequency.

The construction methodology of the dataset uses news websites (e.g. CNN) to inspire human annotators to write multiple choice questions that are provided to models.  Models may use external information (e.g. search engines) to answer the questions and there is no limitation over the information that the models use.

The paper presents sensible set of baselines using open-source methods (e.g. DPR, RAG) and API-based retrieval and inference services (e.g. Google Custom Search, and OpenAI’s GPT LLMs) for comparison. Additionally, a generation benchmark is considered.

The baseline experimentation presents findings considering “time to update” for models against the newly generated questions as part of the investigation. Although i'm not sure if these results can be that valid, they seem a bit unstable  - i'd expect this to be monotonic increasing or decreasing in accuracy. Could this perhaps be due to uncontrollable factors by the 3rd party hosted search or model?

---

> ### Author Response · Authors · 2023-08-11
>
> We are very pleased to know that the reviewer finds our evaluation benchmark exciting and well-positioned.
>
>
> **Updates and Future Plans**
>
> We developed a script for automatic quiz extraction from the websites, thereby mitigating the concern regarding sustainability. We will continually run our benchmark. We also open-source the extraction script and our codebase for all baselines to help researchers who are interested in conducting similar evaluations. We also note that retroactive evaluations with 1500+ examples are also possible. Researchers can evaluate models by controlling access to new documents. Indeed, recent work [1] evaluated their method on our benchmark using this scheme. We will add this discussion point to our manuscript.
>
> [1] Ask Me Anything: A simple strategy for prompting language models. Simran Arora, Avanika Narayan, Mayee F. Chen, Laurel Orr, Neel Guha, Kush Bhatia, Ines Chami, Frederic Sala, and Christopher Ré. In Proceedings of ICLR, 2023.
>
> **Diversity of Source Information**
>
> As noted in our limitations section, we share the reviewer’s concern that topics in RealTime QA can be English-centric (and politics-focused). We will continually look for other source data available on the web to mitigate this problem. As the reviewer points out, however, we believe our benchmark is an important step towards real-world applications of question answering systems.
>
> **Information Change during the Week**
>
> As the reviewer points out, it is possible that there is a discrepancy in accuracy between articles from the start of the week and the end of the week. For this reason, we evaluate all baselines immediately after our announcement time (3 am GMT every Saturday), which is immediately after quizzes are posted on the three websites. We also indicate the submission time of each submission to better understand the performance. Note that there is a trade-off between the availability of information and the temporal gap: the longer a participant waits, the easier it becomes to retrieve recent articles. However, this approach comes at the expense of potential temporal gaps.
>
> **Attribution**
>
> We share the reviewer’s opinion that evaluating the attribution to source information is important for developing trustable AI systems. We make all retrieved documents from DPR and Google custom search available online, facilitating such advanced research explorations.
>
> **Repository Documentation**
>
> Thank you for carefully checking our platform. We will improve our documentation (e.g., more detailed instructions as to how we run evaluation on the latest data or the data from date X through date Y).
>
> **Analysis over Time to Update (Figure 5)**
>
> Thank you for carefully reviewing our manuscript. We have identified a consistent pattern: the performance remains high (or improves) up to around 24 hours after the announcement, but it substantially degrades thereafter. We attribute this pattern to the fact that the 24-hour gap provides access to more recent articles without suffering from the temporal gap. However, as the reviewer has pointed out, it is also plausible that this result stems from the instability of the third-party hosted search. We will incorporate this observation into the revision.

---

### Official Review · Reviewer_bnpu · 2023-07-18
**Real-time evaluation platform for question answering**

**Rating:** 7
**Confidence:** 4
**Correctness:** The claims made in the paper are, as …
**Clarity:** The paper is clear and well-written.

**Strengths:**

This paper introduces a practical proposal for evaluating QA systems on *unseen* information, given that new questions pertaining to current events are used. Given recent concerns over LLMs testing on the training data, this proposal is timely and important.

**Additional Feedback:**

One slight concern is the permanence of the benchmark. The authors do not discuss this, but the Google API used to retrieve supporting information for the weekly releases is not free. To support ~30 questions, the authors may need many searches. The cheapest API tier is not expensive; but the costs can run up. For example, assuming ten pages are retrieved per question, the weekly cost would be a few dollars, and the yearly cost about $100. This is a low estimate. Do the authors have a plan for the continuation of this, perhaps years into the future? This would be good to mention in the paper, also for other groups interested in developing similar benchmarks for other tasks.

**Documentation:**

Sufficient documentation is provided.

**Ethics:**

No ethical concerns (beyond what the authors already discuss in the attached datasheet)

**Limitations:**

The authors include discussion of limitations in the paper.

**Opportunities For Improvement:**

The major problem is the scale: 30 questions/week is a little limited. The limiting factor is the availability of quizzes about current events. However, I would suggest that quizzes are no longer necessary as a primary source of questions: rather, question generation given scraped data from news websites seems plausible. As such, it seems very doable to extend the number of questions served per week to a point where it is not feasible for a malicious human to manually answer them.

One slight improvement to the website that I would suggest is a running average: This would help users better understand the overall performance of models, as opposed to models getting a "lucky break" if events in some week happen to be closer to their training domain.

**Relation To Prior Work:**

Related work is discussed in sufficient detail. If additional space is available a few more sentences expanding the paragraph about prior dynamic benchmarks, lines 290-295, would be appreciated. This benchmark, being a continuous "streaming" platform, is significantly different from the works discussed there; but it is a little unclear (the section just mentions that RealTime is "instantaneous" and "dynamic". I understand the point, but an explanation in the paragraph would be nice).

**Summary And Contributions:**

The paper introduces a platform for real-time evaluation of question answering systems. Every week, ~30 questions pertaining to events in the last week are released. These are decorated with search results from Google. Participants can upload system answers to these, using either the google search results or their own retrieval system. The authors further include two baselines: GPT-3 and RAG. These are discussed and evaluated at length using the platform.

The only significant disadvantage I can spot is that it essentially relies on the honor system: 30 questions per week is small enough that an unscrupulous actor could manually or semi-manually annotate them.

---

> ### Author Response · Authors · 2023-08-11
>
> We are glad to know that the reviewer finds our evaluation scheme timely and important.
>
> **Scale and Maintenance**
>
> We share the reviewer’s opinion that further scaling up our evaluation will strengthen our work and mitigate concerns about malicious actors. We consistently evaluated all baselines in the same setting in the past year, but we plan to extend this work in the near future (e.g., more source websites and question generation from source news data). We note, however, that retroactive evaluations with 1500+ examples are also possible, which leads to larger-scale evaluations. Researchers can evaluate models by controlling access to new documents. Indeed, recent work [1] evaluated their method on our benchmark using this scheme. We will add this discussion point to our manuscript. We also plan to continue to run Google API and pay for the service when necessary using our dedicated research budget.
>
> [1] Ask Me Anything: A simple strategy for prompting language models. Simran Arora, Avanika Narayan, Mayee F. Chen, Laurel Orr, Neel Guha, Kush Bhatia, Ines Chami, Frederic Sala, and Christopher Ré. In Proceedings of ICLR, 2023.
>
>
> **Running Average**
>
> We agree with the reviewer that a running average would be a more robust evaluation of models. We will add this to our website.
>
> **Related Work**
>
> Thank you for checking the manuscript very carefully. We will highlight the distinctive, “streaming” nature of RealTime QA in the related work section of the final version.

---

### Official Review · Reviewer_4qFr · 2023-07-21
**A solid paper but with a dull finding.**

**Rating:** 5
**Confidence:** 4
**Clarity:** Yes, this paper is very easy to under…

**Strengths:**

How QA systems utilize retrieval tools to answer real-time questions is an important research question and a highly valuable application scenario. REALTIME QA has developed an authentic evaluation system and conducted three years of experiments to assess the performance of current state-of-the-art models, which is a robust endeavor. The paper is well-written, presenting a clear overall structure, and all the figures and tables are highly intuitive.

**Additional Feedback:**

N/A

**Correctness:**

I believe the content of this paper is accurate, the experiments are sound, and they can substantiate the authors' claims.

**Documentation:**

This paper publicly discloses the benchmark's URL, reveals the latest questions, and requires participants to submit their system's prediction results via Google Doc, making it essentially reproducible.

**Ethics:**

No, I believe this paper does not include any ethical concerns.

**Limitations:**

The biggest limitation of this paper is that it appears to lack novel discoveries for readers. Although real-time QA is a valuable topic, the difference between real-time QA and traditional document-grounded QA lies only in their data sources. As a result, the authors were unable to draw conclusions that differ significantly from previous works. In the final error analysis, the majority of errors originated from retrieval mistakes, which aligns with the conclusions of prior research.

**Opportunities For Improvement:**

Despite a very solid system design, the authors failed to thoroughly analyze the valuable experimental results they obtained. They merely presented quantitative metrics without providing guidance for future improvement directions. For instance, the authors could have conducted a more comprehensive analysis of the 'Error Breakdown,' particularly regarding retrieval errors and comprehension errors. What would the results be if a stronger retrieval model were used?
Additionally, the weekly evaluation questions for REALTIME QA still require manual annotation and maintenance, which raises concerns about the platform's sustainability in the future.

**Relation To Prior Work:**

I believe that the relevant work in this paper lacks a discussion concerning the previous knowledge-grounded dialogue/QA benchmarks, which assess the performance of a knowledge retrieval module and a knowledge-based question-answering (QA) module. Although this paper focuses on the issue of real-time processing, there is no fundamental difference in their evaluation capabilities.

**Summary And Contributions:**

This paper introduces REALTIME QA, a platform designed for evaluating QA systems' ability to answer factual questions. Each week, REALTIME QA releases questions collected from news websites and evaluates the answers submitted by QA systems through multiple-choice or answer generation. REALTIME QA assesses the performance of models like RAG and GPT-3 in conjunction with DPR and GCS for responding to real-time queries using retrieved information. It also summarizes the potential challenges they may encounter while answering real-time questions.

---

> ### Author Response · Authors · 2023-08-11
>
> We are glad to know that the reviewer finds our evaluation benchmark valuable and our manuscript clear and well structured.
>
> **Novelty and Future Directions**
>
> As the reviewer points out, **if retrieval is correct**, retrieval-augmented, open-book LLMs (Google custom search + GPT-3 in the paper) can accurately answer questions in the RealTime QA benchmark. However, we believe RealTime QA still will give useful guidance for improving LLMs. In real world applications, open-book approaches (i.e., retrieval augmentation) have critical weaknesses: retrieval augmentation results in longer contexts to process, and if we need to reason over many recent events, retrieval approaches become infeasible, and updating outdated LLMs (e.g., finetuning or continual training) would be a more realistic approach. Although we believe this was beyond the scope of this manuscript, RealTime QA has the potential to be used to explore such advanced research questions: for example, how do we remove old information and update LLMs efficiently? We will add this point about the potential usage of RealTime QA in the final version.
>
> **Further Analysis**
>
> We agree with the reviewer that more thorough evaluation would strengthen our work. We hope to illustrate critical mistakes by the examples in Table 3. We will expand Section 3.2 with more detailed analysis.
>
>
> **Maintenance**
>
> We developed a script for automatic quiz extraction from the websites, thereby mitigating the concern regarding sustainability. We will continually run our benchmark. We also open-source the extraction script and our codebase for all baselines to help researchers who are interested in conducting similar evaluations.
>
> **Related Work**
>
> We agree with the reviewer that retrieval-based question answering systems are closely related to knowledge-grounded dialogue or question answering benchmarks. Following the reviewer’s suggestion, we will add a discussion concerning the previous knowledge-grounded dialogue/QA benchmarks (e.g., [1, 2]).
>
> [1] Knowledge-Grounded Dialogue Generation with a Unified Knowledge Representation. Yu Li, Baolin Peng, Yelong Shen, Yi Mao, Lars Liden, Zhou Yu, and Jianfeng Gao. Proceedings of ACL, 2022.
>
> [2] A Survey of Knowledge-enhanced Text Generation. Wenhao Yu, Chenguang Zhu, Zaitang Li, Zhiting Hu, Qingyun Wang, Heng Ji, and Meng Jiang. ACM Computing Surveys, 2022.

---

### Official Review · Reviewer_nqj1 · 2023-07-21
**This work presents an interesting benchmark, RealTimeQA, and a dynamic QA platform**

**Rating:** 7
**Confidence:** 4
**Clarity:** The paper is well written and all not…

**Strengths:**

- This paper proposes initial efforts in resolving real-time QA with time-sensitive questions and provides a dynamic platform for future developments in this direction.

- The authors provide a collection of time-sensitive QA pairs and the latest retrieved source documents from June 17, 2022, to June 2, 2023. Moreover, ~120 QA pairs/month on upcoming time. This can be useful in developing many real-time QA systems.

- Here, a thorough evaluation of six baselines from open-book and close-book QA is conducted and results indicate that the GPT-3 + GCS (open-book system) has better real-time QA capabilities due to access to the most recent articles.

- This work can pave the way for developing more real-world QA systems for many applications.

**Additional Feedback:**

No Additional Feedback

**Correctness:**

The findings of this work are interesting and the experiments are thorough. The dataset is created using an effective framework. However, see the suggestions and questions in the above “Opportunities For Improvement” for more details.

**Documentation:**

Yes, there is sufficient detail on data collection and organization, availability and maintenance, and ethical and responsible use.

**Limitations:**

Yes, the authors adequately addressed the limitations and potential negative societal impact of their work.

**Opportunities For Improvement:**

This is an interesting work in the direction of developing a real-time time-sensitive QA system. However, there are a few questions and suggestions to improve the work -

1. Both recent systems, Microsoft Bing and Google BARD, have the capability to search the internet and generate responses for users. Additionally, these systems have access to a vast and diverse set of information from the internet, extending beyond just news articles. In comparison, could you please explain how your work distinguishes itself from these existing systems?

2. I think it is important to add systems like BARD or Bing as baselines and see the performance on time-sensitive QA. Have you evaluated these systems on your collected data?

3. I have a question about the annotation framework - How do you make sure of gold answer sets for the questions that you extracted?

4. For the document retrieval, why only explores top-5 documents for answering the question? It is good to present an ablation study for different top-k retrieved documents, where k can range from 1<k<n.

**Relation To Prior Work:**

Yes, it clearly discussed how this work differs from previous contributions. Please add discussion related to existing systems such as BARD and Microsoft Bing.

**Summary And Contributions:**

This paper presents RealTimeQA, a dynamic QA platform to fulfill real-time, instantaneous information to answer time-sensitive questions. This work develops an annotation framework and benchmarking timeline to achieve this and provides a dataset with time-sensitive QA pairs with up-to-date source documents. GCS API is used to extract real-time documents from newly-published news articles spanning various genres. This work differs from previous annotated QA datasets which rely on information available at the time of data creation. Six different open-book and close-book QA baselines are evaluated on a regular basis on RealTimeQA. Results indicate that GPT-3 (open-book) + GCS outperforms all baselines exhibiting better real-time QA capabilities. RealTimeQA can be beneficial for many real-world applications.

---

> ### Author Response · Authors · 2023-08-11
>
> We are happy to hear that the reviewer found our analysis is thorough and our evaluation platform useful for future development of question answering models.
>
> **Additional Baselines**
>
> We thank the reviewer for the suggestion. We are indeed considering adding more baselines to our platform: retrieval-augmented large language models, such as BARD and Bing. At the time of starting our benchmark, these models were not available, so we focused on year-long comparisons of our baselines in this paper. Since RealTime QA is dynamic, it is possible to incorporate new baselines. We will add preliminary results of new baselines in the final version.
>
> **Annotations**
>
> We have gold answers from the multiple-choice questions from the three news websites. In the generation setting, we only have one gold answer per question. As the reviewer points out, it is ideal to cover multiple gold answers in the generation setting (vs. the multiple-choice setting) so that surface-level differences will be disregarded (e.g., President Joe Biden vs. Joe Biden). To mitigate this issue, we follow previous work and use the F1 score in addition to exact matching. We clarify this point in the final version.
>
> **Analysis with Varying Numbers of Retrieved Documents**
>
> We thank the reviewer for the suggestion to improve our work. We will add a plot that compares the performances with varying numbers of retrieved documents in the final version.

---

### Official Review · Reviewer_SxTx · 2023-07-23
**Review of RealTime QA**

**Rating:** 5
**Confidence:** 4
**Correctness:** Yes.
**Clarity:** Yes.

**Strengths:**

The evalution of realtime question answering is important and this paper introduces a dynamic platform that announces questions and evaluates systems every week. Based on the collected questions, performance over varying submission time are also analysized.

**Additional Feedback:**

None.

**Documentation:**

Yes.

**Ethics:**

None.

**Limitations:**

The details of choice extraction can be more clear in the paper.

**Opportunities For Improvement:**

The main issue I concern is that the raw data of evaluation are limited to three websites, which may be unstable, and the curation depends on human efforts. In addtion, the results on github (https://github.com/realtimeqa/realtimeqa_public/tree/main/latest) show that the latest update is on 2023-06-09, and a more timely and automatic update is needed.

**Relation To Prior Work:**

Yes.

**Summary And Contributions:**

This paper focuses on an intereting and important problem: the evaluation of realtime question answering, i.e., answering questions about novel events or information. A platform for such evaluation is made public, by regularly collecting questions from three websites:  CNN, USA Today, and The WEEK, extracting choices, and retrieving the corresponding document corpora. 6 models are taken as baselines for fair evaluation, including from open-book models, closed-book models, prompting-based models, and finetuning-based models.

---

> ### Author Response · Authors · 2023-08-11
>
> We are glad to hear that the reviewer finds our dynamic platform for question answering important.
>
> **Limitations**
>
> As noted in the limitations section, we agree with the reviewer that adding more resources beyond the three news websites will strengthen our platform. We are currently working to add more resources, though many of these websites require subscriptions and make it difficult to extract quizzes automatically. We note, however, that retroactive evaluations with 1500+ examples are also possible: practitioners can evaluate models by controlling access to new documents. Indeed, recent work [1] evaluated their method on our benchmark using this scheme. We will add this discussion point to our manuscript.
>
>
> [1] Ask Me Anything: A simple strategy for prompting language models. Simran Arora, Avanika Narayan, Mayee F. Chen, Laurel Orr, Neel Guha, Kush Bhatia, Ines Chami, Frederic Sala, and Christopher Ré. In Proceedings of ICLR, 2023.
>
> **Updates**
>
> Thank you for carefully checking our platform. We did not update the results on our public github repository during the review period for clarity. We updated the public repository with our results after submission of this manuscript.
>
> **Choice Extraction**
>
> We use an automatic script for choice extraction from the three websites. We automatically download the quiz data and parse it into a set of questions with multiple choices. We will elaborate on this in the final version of our manuscript.

---

### Author Response · Authors · 2023-08-11
**General response to all reviews**

We are pleased to learn that all five reviewers recognize the significance and value of our evaluation efforts in developing time-sensitive systems. We would like to highlight that our RealTime QA benchmark has also been used for retroactive model evaluations, encompassing over 1500 questions [1]. This illustrates its utility within the research community beyond weekly evaluations. We address questions, suggestions, and concerns point by point for each reviewer.

[1] Ask Me Anything: A simple strategy for prompting language models. Simran Arora, Avanika Narayan, Mayee F. Chen, Laurel Orr, Neel Guha, Kush Bhatia, Ines Chami, Frederic Sala, and Christopher Ré. In Proceedings of ICLR, 2023.

---

### Decision · Program_Chairs · 2023-09-22

**Decision:**

Accept (Poster)

**Comment:**

This paper introduces a platform for real-time question answering evaluation. Unlike previous approaches that treat question answering as a static task, this platform provides a dynamic dataset that updates weekly. This platform can facilitate the development of online QA systems that can handle time-sensitive questions. Although, there are a few concerns: 1) the challenges of model evaluation; 2) the absence of commercial system benchmarks such as Microsoft Bing and Google BARD. Overall, the paper makes a valuable contribution to the field of real-time question answering.